# Electrolytes with Micelle-Assisted Formation of Directional Ion Transport Channels for Aqueous Rechargeable Batteries with Impressive Performance

**DOI:** 10.3390/nano12111920

**Published:** 2022-06-04

**Authors:** Yanmin Lu, Fengxiang Zhang, Xifeng Lu, Haihui Jiang, Wei Hu, Libin Liu, Ligang Gai

**Affiliations:** 1Engineering & Technology Center of Electrochemistry, School of Chemistry and Chemical Engineering, Qilu University of Technology, Shandong Academy of Sciences, Jinan 250353, China; 17854116233@163.com (Y.L.); 1043119202@stu.qlu.edu.cn (F.Z.); jhh@qlu.edu.cn (H.J.); lbliu@qlu.edu.cn (L.L.); 2School of Energy Materials, Shandong Polytechnic College, Jining 172000, China; lxf-1979@126.com

**Keywords:** sodium dodecyl sulfate, micelle cluster, directional ion transport channel, electrolyte, aqueous rechargeable battery

## Abstract

Low-cost and ecofriendly electrolytes with suppressed water reactivity and raised ionic conductivity are desirable for aqueous rechargeable batteries because it is a dilemma to decrease the water reactivity and increase the ionic conductivity at the same time. In this paper, Li_2_SO_4_–Na_2_SO_4_–sodium dodecyl sulfate (LN-SDS)-based aqueous electrolytes are designed, where: (i) Na^+^ ions dissociated from SDS increase the charge carrier concentration, (ii) DS^−^/SO_4_^2−^ anions and Li^+^/Na^+^ cations are capable of trapping water molecules through hydrogen bonding and/or hydration, resulting in a lowered melting point, (iii) Li^+^ ions reduce the Krafft temperature of LN-SDS, (iv) Na^+^ and SO_4_^2−^ ions increase the low-temperature electrolyte ionic conductivity, and (v) SDS micelle clusters are orderly aggregated to form directional ion transport channels, enabling the formation of quasi-continuous ion flows without (r.t.) and with (≤0 °C) applying voltage. The screened LN-SDS is featured with suppressed water reactivity and high ionic conductivity at temperatures ranging from room temperature to −15 °C. Additionally, NaTi_2_(PO_4_)_3_‖LiMn_2_O_4_ batteries operating with LN-SDS manifest impressive electrochemical performance at both room temperature and −15 °C, especially the cycling stability and low-temperature performance.

## 1. Introduction

Aqueous rechargeable metal-ion batteries (ARMBs) are perceived as promising alternatives that can settle the major challenges facing conventional lithium-ion batteries, because water serving as an environment-benign solvent yields a lowered fabrication cost, improved rate capability, and exempt safety concerns [1,2]. However, ARMBs operating with dilute electrolytes usually suffer from inferior cycling stability, due to the dissolution of the electrodes and side reactions associated with water, oxygen, and/or solute in the electrolyte [3,4,5]. For example, LiMn_2_O_4_ (LMO) has been widely used as a cathode material for “rocking-chair” aqueous rechargeable lithium-ion batteries (ARLBs) [6,7], due to its redox potential (vs. Li/Li^+^) suitable for batteries operating with aqueous electrolytes [1,8]. However, delithiated LMO is proven to be catalytic upon oxygen evolution reaction (OER), meaning promoted OER on the cathode and a lowered upper voltage limit of the electrolyte and, hence, deteriorated battery performance [9].

To improve the cycling stability of ARMBs, many strategies have been designed, such as using halogen conversion–intercalation chemistry [10], cathode design and modification [11,12,13], decoupling electrolytes [14], adding an interface-forming additive [5,15], eliminating O_2_ and adjusting pH [3,16], and reducing the amount of free water molecules in electrolytes [8,17,18,19,20,21,22,23,24,25,26,27,28]. Recently, a type of “water-in-salt” (WIS) electrolyte has been demonstrated to be effective in improving the specific energy and cycling stability of ARMBs [29], due to the reduced water reactivity and the elevated output voltage. However, the use of high-concentration and fluorinated salts, such as lithium bis(trifluoromethane)sulfonimide (LiTFSI) [17], lithium bis(pentafluoroethane)sulfonamide (LiBETI) [18], and lithium trifluoromethanesulfonate (LiOTf) [19], causes toxicity and cost concerns which impede their commercialization [29]. To mitigate these issues, “molecular-crowding” aqueous electrolytes [8], “water-in-ionomer”/“water-in-polymer salt” electrolytes (WIPSE) [27,30,31], localized WIS electrolytes [28], and low-cost alkali metal-ion nitrate [20], sulfate [20], acetate [25], and formate [26] with saturated concentrations in water have been developed to improve the ARMB performance. Although ARMBs operating with concentrated or diol-added electrolytes manifest enhanced specific energy and cycling stability [8,17,18,19,20,21,22,23,24,25,26,27,28], their rate capability is compromised due to their relatively low ionic conductivities (Table 1). Therefore, it is highly desirable to find an electrolyte that has both suppressed water reactivity and high ionic conductivity.

As an anion surfactant, sodium dodecyl sulfate (SDS) has been widely used in the petroleum industry, environmental remediation, pharmaceutics, and daily chemicals [33,34]. Additionally, there has been an increasing interest in using SDS as an electrolyte additive for energy storage systems [35,36,37]. For example, the dendrite formation and corrosion on Zn anodes can be suppressed by electroplating Zn foils in an aqueous electrolyte containing trace amounts of SDS, presenting Zn‖LiFePO_4_ [36] and Zn‖LMO [37] batteries with enhanced cycling stability. Huang and co-workers developed a polyethylene glycol (PEG)-SDS-H_2_O electrolyte for 2.5 V symmetric supercapacitors [35], in which PEG serves as a water-trapping agent and SDS as a salt to afford ionic conductivity. Hou and co-workers found that an electrolyte with an electrochemical stability window (ESW) of ca. 2.5 V can be obtained by adding 1 cmc (critical micelle concentration) SDS into Na^+^/Zn^2+^-mixed aqueous electrolytes [21].

As documented in the literature, (i) the conductivity of SDS aqueous solution is enhanced with the presence of metal ions, due to the increase in the charge carrier concentration [38], (ii) the number of water molecules associated with each DS^−^ is eight [39], and (iii) the addition of Li^+^ salt into SDS solution can reduce the Krafft temperature (*T*_K_), above which the conductivity suddenly rises due to the increase in surfactant solubility [40,41]. These results suggest that an electrolyte with high ionic conductivity, reduced water reactivity, and a widened operating temperature range can be achieved for ARLBs, in a case where concentrated SDS-based electrolytes containing Li^+^ ions were employed.

In this context, we designed a new type of electrolyte in terms of high-concentration SDS in Li^+^/Na^+^-mixed aqueous solution, hereafter referred to as LN-SDS-n, where n denotes the number of multiple cmcs (1 cmc = 8.4 mM, r.t.) [42]. After comparative investigation on the physicochemical properties of LN-SDS-n (1 ≤ n ≤ 120), NaTi_2_(PO_4_)_3_(NTP)‖LMO hybrid metal-ion batteries were taken as a probe to evaluate the electrolyte performance. A Na^+^ insertion/extraction anode coupling with a Li^+^ insertion/extraction cathode could enrich the ARMB community [43,44]. As expected, LN-SDS-90 endows NTP‖LMO with impressive electrochemical performance.

## 2. Experimental

### 2.1. Materials

LMO was purchased from Jining Wujie Science and Technology Ltd., Jining, China. Waterborne polyurethane was obtained from SiwoChem Ltd., Shanghai, China. Analytical reagents of sodium hydroxide (NaOH), phosphoric acid (H_3_PO_4_), sodium sulfate (Na_2_SO_4_), lithium sulfate (Li_2_SO_4_), anhydrous ethanol, 1-butanol, tetrabutyl titanate (TBT), *N*-methyl-pyrrolidone (NMP), and SDS were used as-received without further purification.

### 2.2. Preparation of NTP

NTP was prepared through a modified method reported in [45]. Briefly, 3 mmol NaOH and 9 mmol H_3_PO_4_ were dissolved in 5 mL of distilled water to form solution A, and 6 mmol TBT was dissolved in a mixed solution containing 25 mL of anhydrous ethanol and 25 mL of 1-butanol to form solution B. Then, solution A was slowly dropped into solution B with agitation at room temperature for 2 h, followed by dropwise addition of 40 mL of 5 wt.% waterborne polyurethane with vigorous stirring for 2 h. The precipitate was filtered, washed with distilled water several times, and finally dried in a vacuum oven at 60 °C to obtain the precursor. NTP powders were obtained by annealing the precursor in nitrogen atmosphere at 700 °C for 2 h with a heating rate of 5 °C min^−1^ and allowed to cool to room temperature.

### 2.3. Preparation of LN-SDS-n Electrolytes

First, a 0.5 mol L^−1^ Li_2_SO_4_–0.5 mol L^−1^ Na_2_SO_4_ mixed solution was prepared, followed by the addition of a given amount of SDS. The mixture was stirred at room temperature for 10 min, sealed, and then statically kept at 25 °C to produce pellucid LN-SDS-n electrolytes without any bubbles.

### 2.4. Characterization

X-ray powder diffraction (XRD) spectra were recorded on a Bruker D8 Advance diffractometer (Bruker Corporation, Karlsruhe, Germany) with Cu Kα radiation (*λ* = 1.5406 Å). Measurements by scanning electron microscopy (SEM) were performed on a Hitachi Regulus8220 field-emission scanning electron microscope (Hitachi, Ltd., Tokyo, Japan). Measurements by transmission electron microscopy (TEM) were conducted with a JEOL JEM-2100 high-resolution transmission electron microscope (JEOL, Akishima-shi, Japan). Room-temperature Raman spectra were recorded on a Renishaw inVia plus laser Raman spectrometer (Renishaw, Gloucestershire, UK) with resolution of 2 cm^−1^ under laser excitation at 532 nm. Measurements by differential scanning calorimetry (DSC) were conducted on a TA Instruments DSC2500 differential scanning calorimeter (TA Instruments, New Castle, DE, USA). To eliminate the effect of thermal history resulting from storage temperature and time [46], sample electrolytes (10–15 mg) were sealed and heated to 40 °C from room temperature, followed by cooling to −50 °C, keeping at −50 °C for 5 min, and then heating to 40 °C, with a ramp rate of 10 °C min^−1^. Rheological measurements were performed on a TA DHR-2 interfacial rheometer (TA Instruments, New Castle, DE, USA), using a parallel plate with a diameter of 25 mm. First, a dynamic strain scan at a 10 r s^−1^ angular frequency in the range of 0.1–100% was used to determine the linear viscoelastic region. The frequency scan was then performed at frequencies of 0.1–100 r s^−1^ in the linear viscoelastic region, using a strain fixation of 1%. Measurements by *ζ*-potential of sample electrolytes were conducted on a Mastersizer 2000 laser size detector (Malvern Instruments Limited, Malvern, UK).

### 2.5. Electrochemical Test

Cyclic voltammograms (CVs), galvanostatic charge/discharge (GCD) curves, electrochemical impedance spectra (EIS), and linear sweep voltammetry (LSV) curves for the electrodes, electrolytes, and batteries were recorded on a CHI760E electrochemical workstation (Shanghai CH Instruments Co., Shanghai, China) and/or a LANHE CT2001A battery tester (Wuhan Landian Co., Wuhan, China). For EIS measurements, the ac amplitude was set at 5 mV and the frequency was in the range of 0.01–10^6^ Hz.

LMO and NTP electrodes were prepared by coating slurries, made from blending the active material, acetylene black, and polyvinylidene fluoride (8:1:1, weight ratio) in N-methyl pyrrolidone, onto strips of stainless-steel cloth with a coating area of 1 × 1 cm, followed by drying in a vacuum oven at 80 °C for 12 h. Individual anodes and cathodes with active material of 2–3 mg cm^−2^ were first evaluated through a three-electrode system, using an Ag/AgCl reference electrode, a platinum foil counter electrode, and LN-SDS-0 as the electrolyte. Aqueous NTP‖LMO batteries were prepared by assembling R2032-type coin cells in an ambient environment, using Whatman^TM^ GF/A glass microfiber filters (GE Healthcare Life Sciences, Beijing, China) as the separator and LN-SDS-n as the electrolyte, where n equals 0, 1, 30, 60, 90, and 120, respectively. The cathode material is 2.3–3.5 mg cm^−2^ relative to the anode material of 2–3 mg cm^−2^, based on the anode/cathode capacity ratio of ca. 13/12. The anode/cathode capacity ratio is larger than 1, due to the larger anode polarization in batteries and the high activity of NTP in the Na-intercalation state [47,48,49].

The electrochemical stability window of LN-SDS-n electrolytes was determined by LSV in a three-electrode system, using graphene-coated aluminum foil (Kunshan Boerfa New Materials, Ltd., Kunshan, China) with a diameter of 19 mm as the working and counter electrodes, and Ag/AgCl electrode as the reference. The reason for selecting graphene-coated aluminum foil as the current collector was to obtain the electrolyte ESW in a more realistic environment [8], as high-surface-area graphene has been frequently used as an electrode material for LIBs [50].

### 2.6. Computational Method

All dynamic simulations were performed by using the GROMACS software package (GROMACS5.1.4, Groningen, The Netherlands) [51,52,53], in which the gromos54a7_atb forcefield was used for all components [54]. The SPC model and TIP4P/Ice model were used to describe H_2_O molecules in liquid (water) and solid (ice) states [55,56]. The interactions for all other atomic species were modeled by using Lennard–Jones potential with Lorentz–Berthelot combination rules. Electrostatic and van der Waals interactions were calculated through the PME and truncation methods, respectively [57]. The simulation steps consist of: (1) a minimized energy system, (2) a production run of 4 ns without an electric field (NPT ensemble), and (3) a production run of 5–30 ns with an electric field to reach the equilibrium state (NPT ensemble).

The sizes of the boxes with SDS (100 SDS, 7351 H_2_O, 132 Na^+^, 132 Li^+^, and 132 SO_4_^2−^) and without SDS (7351 H_2_O, 132 Na^+^, 132 Li^+^, and 132 SO_4_^2−^) were set at 64.0 Å × 64.0 Å × 64.0 Å and 60.0 Å × 60.0 Å × 60.0 Å in sequence, using three-dimensional periodic boundary conditions. A series of temperature (298.15, 273.15, 268.15, 263.15, and 258.15 K) and electric field parameters (0.5, 1.0, 1.5, and 2.0 V nm^−1^) were considered in the present simulations. To understand the electrolyte behavior in different conditions, we calculated mean squared displacement (MSD) and the number of hydrogen bonds of water from the dynamic trajectory.

## 3. Results and Discussion

### 3.1. LN-SDS-n Electrolytes

Figure 1a shows the LSV curves of LN-SDS-n at room temperature. The ESWs of LN-SDS-n (n ≥ 1) are in the range of 1.78–2.06 V vs. RHE (reversible hydrogen electrode), larger than that of 1.72 V for LN-SDS-0. The ESWs of LN-SDS-n increase as n increases to 90 (Figure 1a, inset). This result suggests that highly concentrated SDS facilitates to reduce the water reactivity, due to the formation of hydrogen bonding between water and SDS head groups [58]. However, the ESW of LN-SDS-120 is lower than that of LN-SDS-90, meaning a minimized water reactivity for LN-SDS-90 [8]. The abnormal electrochemical behavior of LN-SDS-n can be explained by how the orientational dynamics of water molecules and the hydrogen bonding between water and SDS head groups are not linearly dependent on the surfactant concentration [58,59].

To further reveal the electrochemical stability of LN-SDS-n, room-temperature Raman spectra were collected as shown in Figure 1b. A dramatic increase in *v*_as_(CH_3_,CH_2_) and *v*_s_(CH_3_,CH_2_) signals (2800–3000 cm^−1^) occurs as n ≥ 30, showing the existence of SDS [60]. However, the *v*(OH) band (3000–3800 cm^−1^) decreases in intensity as n increases to 90 and then increases at n equals 120. This result suggests a minimized water reactivity for LN-SDS-90 [34], in concert with the ESW result (Figure 1a, inset).

The room-temperature ionic conductivity of LN-SDS-n was measured by the EIS technique, as shown in Appendix A. The conductivity of LN-SDS-n (n ≥ 1) is in the range of 41–93 mS cm^−1^ (Figure 1c), much higher than that of LN-SDS-0, organic carbonate-based Li^+^ electrolytes [59], LiTFSI-based WIS [8,17], water-in-bisalt (WIBS) [18], and LiNO_3_(KOAc, LiOAc)-based WIS [25,28], and comparable to that of 10–40 m HCOOK and potassium polyacrylate and derivatives (Table 1) [26,30,31]. As mentioned before, the coexistence of SDS with metal ions helps to increase the electrolyte conductivity, due to the improved charge carrier concentration [37]. In the present case, SDS micelles are formed in LN-SDS-n (n ≥ 1). The conductivity is solely due to the free counter-cations from SDS and added salts (Li_2_SO_4_ and Na_2_SO_4_). The micelles indirectly contribute to the conductivity through dissociated Na^+^ from the micelle surface [61]. Although LN-SDS-120 provides more Na^+^ ions compared with LN-SDS-90, the latter exhibits the highest conductivity.

To interpret this result, the rheological properties of LN-SDS-n (n = 30–120) were examined (Figure 1d–f). Figure 1d shows the shear viscosity (*η*) plots as a function of shear rate ($\dot{\gamma }$). For LN-SDS-30, the viscosity is nearly independent of the shear rate, showing a Newtonian behavior [62]. As for LN-SDS-n (n = 60–120), there is a Newtonian plateau at low shear rates followed by a decrease in viscosity at higher shear rates, exhibiting a shear thinning behavior [62]. Figure 1e,f show the extensional rheological characteristics of LN-SDS-n (n = 30–120), where Gʹ, Gʺ, *η**, and *ω* are the elastic (or storage) modulus, viscous (or loss) modulus, complex dynamic viscosity, and angular frequency in sequence. At low frequencies, Gʺ is located above Gʹ, especially for LN-SDS-n (n = 30–90), suggesting a fluid-like behavior. At high frequencies, Gʹ is located well above Gʺ for LN-SDS-n (n = 30–120), suggesting a solid-like behavior (Figure 1e) [62]. Additionally, *η** decreases with increasing *ω* (Figure 1f). According to [62], the extensional rheological results indicate that LN-SDS-n (n = 30–120) belongs to the entangled solution. Note that the intersection point position of Gʹ and Gʺ decreases as n increases (Figure 1e), due to the increasing viscosity with increasing SDS concentration.

In brief, the rheological results of LN-SDS-n (n = 30–120) show that both zero shear viscosity and *η** increase as n increases (Figure 1d,f). The increasing viscosity yields an increasing concentration of metal ions immobilized on the micelle surface, as reflected by the *ζ*-potential plot, where the *ζ*-potential values change in a positive direction (Figure 1g). Meanwhile, the increasing viscosity is adverse to ion migration. As n equals 120, the negative effect of increasing viscosity surpasses the positive contribution of increasing ion concentration, leading to a lowered conductivity compared with LN-SDS-90.

Water-based electrolytes with anti-freezable ability and acceptable ionic conductivity at subzero temperatures remain a big challenge [63,64]. To examine the anti-freezable performance of LN-SDS-n, DSC measurements were performed, as shown in Figure 2a. The melting point decreases as n increases to 30 and then increases with increasing n from 30 to 120 (Figure 2a, inset), but all the melting points for LN-SDS-n (n ≥ 1) are lower than those for LN-SDS-0. This result indicates that SDS plays a role in lowering the melting point of LN-SDS-n (n ≥ 1). On the other hand, Li^+^/Na^+^ ions resulting from the added inorganic salts contribute to lower the melting point. For example, LN-SDS-90 exhibits a melting point of −3.71 °C, much lower than that of −0.35 °C for SDS-90 (Figure 2a), which is a SDS aqueous solution without adding any inorganic salts. Therefore, the lowered melting point for LN-SDS-n (n ≥ 1) is due to the cooperative interactions of SDS with Li^+^/Na^+^ and SO_4_^2−^ ions. As documented in the literature, DS^−^/SO_4_^2−^ anions and Li^+^/Na^+^ cations are capable of trapping water molecules through hydrogen bonding [57] and/or hydration [61], respectively. As a result, the hydrogen bonding between water molecules may be disturbed, leading to the increase in non-freezable water concentration of LN-SDS-n (n ≥ 1) [63].

The non-freezable water is generated through the interaction of water with other components in the system and does not show a phase transition in calorimetric analysis [63]. To interpret the DSC result, relative concentrations of freezable water in LN-SDS-n were calculated according to Equation (1), expressed as [63,64]:(1)$$W_{f}=\frac{\Delta H_{m}}{\Delta H_{m}^{{}^{\circ}}W_{H_{2}O}}$$
where *W*_f_ and Δ*H*_m_ are the weight percent and melting enthalpy of freezable water in sequence, $\Delta H_{m}^{{}^{\circ}}$ is the melting enthalpy of pure water with a value of 333.5 J g^−1^, and $W_{H_{2}O}$ is the relative concentration of water in the electrolyte, $W_{H_{2}O}=m_{H_{2}O}/m_{\mathrm{total}}$. The calculation results (Figure 2b) show that the changing rule in *W*_f_ is consistent with that in the melting point (Figure 2a, inset), i.e., the lower the freezable water concentration in LN-SDS-n (n = 0–120), the lower melting point it has.

Apart from the melting peak, a weak endothermic peak occurred for LN-SDS-n (n = 30–120). The weak peak position is centered at 7.6, 8.9, 10.9, and 11.3 °C as n equals 30, 60, 90, and 120, respectively (Figure 2a). The weak peak is tentatively assigned to the *T*_K_, as confirmed by the conductivity plot as a function of temperature (Figure 2c). As n equals 30, 60, 90, and 120, the *T*_K_ was measured to be ca. 3.6, 3.9, 9.9, and 10.2 °C in sequence, much lower than that of 16 °C for SDS in concentrated SDS–H_2_O systems without foreign metal ions [65,66]. Likewise, the weak peak position for LN-SDS-90 (10.9 °C) was much lower than that of 18.1 °C for SDS-90 (Figure 2a). The abrupt increase in the plots of ionic conductivity vs. temperature is characteristic of surfactant solutions [39,40]. The lowered *T*_K_ for LN-SDS-n is attributed to the Li^+^ ions, which are capable of reducing the *T*_K_ of SDS-based solutions [39,40]. Note that the lowered *T*_K_ for LN-SDS-n is helpful for improving the electrolyte conductivity at low temperatures.

The conductivity for conventional WIS-based electrolytes dramatically declined as the temperature deceased [17,28]. For example, the conductivity of 25 m LiNO_3_ decreased to only 0.016 mS cm^−1^ at 10 °C from 73.8 mS cm^−1^ at room temperature [28]. The conductivity of the well-known 21 m LiTFSI is only 2.4 mS cm^−1^ at 0 °C (Table 2). As for LN-SDS-n (n = 30–120), high conductivity values can be achieved even at temperatures below the melting points (Table 2). For example, the conductivity for LN-SDS-n (n = 30, 60, 90, 120) at −15 °C was 12.6, 15.5, 16.9, and 14.0 mS cm^−1^ in sequence, much higher than that of 1.6 mS cm^−1^ for LN-SDS-0 and 1.0 mS cm^−1^ for 21 m LiTFSI, and higher than the room-temperature conductivity of WIS (3–10 mS cm^−1^) [17,18,25] and organic carbonate-based Li^+^ electrolytes (4.5–10.7 mS cm^−1^) [59].

The relatively high ionic conductivity of LN-SDS-n (n ≥ 30) at low temperatures can be deeply understood by the Arrhenius equation, expressed as [63,67]:*σ* = *σ*_0_ exp(−*E*_a_/*kT*)
where *σ* is the ionic conductivity, *σ*_0_ is the pre-exponential factor, *E*_a_ is the activation energy, *k* is the Boltzmann’s constant, and *T* is the absolute temperature.

Figure 2d shows the plots of ionic conductivity vs. the reciprocal of absolute temperature. To eliminate the effect of *T*_K_ above which the ionic conductivity abruptly increases, we analyzed the plots in the temperature range of −15–5 °C. A linear relationship exists between log *σ* and *T*^−1^ for LN-SDS-n (n = 30–120), indicating that the electrolyte conductivities obey the Arrhenius law at the measured temperatures [67]. The Arrhenius relationship means the ion motion is liquid-like [67]. In other words, LN-SDS-n (n = 30–120) cannot be completely solidified even at −15 °C. For LN-SDS-0, however, the linear relationship exists only at the temperatures above −10 °C and an inflection point occurs after that. This result suggests that LN-SDS-0 cannot work at temperatures below −10 °C, due to solidification.

For LN-SDS-n (n = 30–120), the *E*_a_ decreased as n increased from 0 to 90, and then increased with increasing n to 120 (Table 2). The *E*_a_ is an energy barrier that must be overcome for ion transfer. A lower *E*_a_ means easier ion migration of an electrolyte [68]. Therefore, the higher ionic conductivity of LN-SDS-n (n = 30–120) compared with LN-SDS-0 and 21 m LITFSI is attributed to the lowered *E*_a_, which results from the liquid-like ion motion due to the anti-freezable ability.

### 3.2. Conductive Mechanism of LN-SDS-n

Recently, our research group has designed a zwitterionic polymer hydrogel electrolyte with an outstanding conductivity of 12.6 mS cm^−1^ at −40 °C [63]. The high conductivity is attributed to the hopping migration of hydrated Li^+^ through the channel of zwitterion groups, which function as dissociation enhancers for lithium salts [69,70]. For LN-SDS-n, high conductivity values can be obtained at temperatures below the melting points (Table 2). This is attributed to the lowered activation energy for ion migration due to the anti-freezable ability. To further interpret the abnormal electrochemical behavior of LN-SDS-n, theoretical computations were performed (Figure 3). In the case of LN-SDS-90 where no electric field was applied (Appendix A), Li^+^, Na^+^, SO_4_^2−^, and SDS were randomly distributed at the temperatures ≤ 0 °C (273.15–258.15 K). Such a situation also occurred in LN-SDS-0 (Appendix A). At room temperature (298.15 K), Li^+^, Na^+^, and SO_4_^2−^ tend to reside upon the SDS molecular clusters, leading to the formation of a directional ion transport channel (Appendix A). For LN-SDS-0 at room temperature, however, ion clusters were formed and randomly distributed, without forming directional ion transport channels (Appendix A).

When applying an electric field with a voltage of 0.5–2.0 V nm^−1^ on room-temperature LN-SDS-90, the ions upon the SDS clusters tend to move into the ion transport channels (Appendix A). Of more importance, SDS molecules are orderly aggregated for low-temperature (273.15–258.15 K) LN-SDS-90 when applying a voltage of 2.0 V nm^−1^, resulting in the formation of quasi-continuous ion flows in ion transport channels (Figure 3). It should be noted that the formation of directional ion transport channels at low temperatures becomes more apparent as the applied voltage increases (Appendix A). This result means the positive function of applied voltage on the ion transport. For LN-SDS-0, however, discontinuous ion flows dominate the ion transport at temperatures ≤0 °C (273.15–258.15 K), even when applying a high voltage of 2 V nm^−1^ (Appendix A). Quasi-continuous ion flow only occurs for room-temperature LN-SDS-0 with applying voltage (Appendix A).

The SDS micelle-assisted formation of directional ion transport channels facilitated the improvement of the ionic conductivity, as reflected by the MSD plots (Figure 4a–f). The presence of SDS can improve the room-temperature MSD of Li^+^, Na^+^, and SO_4_^2−^ species, especially for the case where a high voltage of 2 V nm^−1^ is applied. At low temperatures (273.15–258.15 K), the Li^+^ MSD values for LN-SDS-90 were not improved in the applied voltage range of 0.5–2.0 V nm^−1^, as compared to LN-SDS-0. However, the Na^+^ and SO_4_^2−^ MSD values for LN-SDS-90 greatly improved, especially for the case with applied voltages of more than 1.0 V nm^−1^. The MSD results indicate that the presence of SDS provides great contributions to improve the Na^+^ and SO_4_^2−^ migration rates, but has a minor effect on the Li^+^ migration rate. In other words, the improved ionic conductivity for LN-SDS-90 at the temperatures ≤ 0 °C is mainly attributed to the improved Na^+^ and SO_4_^2−^ migration rates.

As previously mentioned, the hydrogen bonding between water molecules may be disturbed by the foreign species such as SDS, Li_2_SO_4_, and Na_2_SO_4_, resulting in a decrease in the freezable water concentration and, hence, lowered electrolyte melting points (Figure 2a,b). In this regard, hydrogen bond numbers (HBNs) of water in electrolytes have also been computed (Figure 4g,h). The HBNs of water in electrolytes increased as the temperature decreased. Note that the HBNs for LN-SDS-90 were lower than those for LN-SDS-0 at the temperatures ranging from 298.15 to 258.15 K (Table 3). The lowered HBNs for LN-SDS-90 mean a reduced freezable water concentration as compared with LN-SDS-0. In this regard, the theoretical computations support the DSC result.

Based on the above analyses, the high ionic conductivity of LN-SDS-90 in a wide temperature range of 298.15–258.15 K (−15–0 °C) is attributed to the formation of directional ion transport channels without (at 298.15 K) and with (at 273.15–258.15 K) applying an electric field. When working, the cations and anions form quasi-continuous ion flows in the channels, enabling the high ionic conductivities. In addition, the decreased HBNs of water, reduced freezable water concentration, and improved Na^+^ and SO_4_^2−^ migration rates are helpful for improving the ionic conductivity of LN-SDS-90 at temperatures below the melting point. The high ionic conductivities in a wide temperature range make LN-SDS-90 a promising electrolyte for ARMBs.

### 3.3. NTP‖LMO Operating at Room Temperature

To put LN-SDS-n in real applications, NTP‖LMO batteries were taken as a probe to evaluate the electrolyte performance. Before battery assembly, structural characterization and electrochemical tests for individual anodes and cathodes (Appendix A) were performed in a three-electrode system, using LN-SDS-0 as the electrolyte. Corresponding results and discussions are provided in the Appendix A. In brief, (i) the self-made anode material is a carbon-coated NTP with a carbon layer of 2–6 nm and particle sizes in the range of 60–120 nm (Appendix A), (ii) the discharge-specific capacity (*C*_dis_) of NTP retained 56 mA h g^−1^ after 1000 cycles at 1 A g^−1^, presenting a capacity retention of 64.6% relative to the initial *C*_dis_ (Appendix A), and (iii) the commercial LMO retained only 8 mA h g^−1^ after 1000 cycles at 1 A g^−1^, showing an inferior cycling stability (Appendix A). Therefore, it remains a formidable task to improve the cycling stability of NTP‖LMO batteries.

Figure 5 shows the electrochemical performance of NTP‖LMO operating with LN-SDS-n. As n equals 0, 1, 30, 60, 90, and 120, the NTP‖LMO full cells at 0.2 C (1 C = 133 mA g^−1^) delivered *C*_dis_ values of 59.4, 67.1, 62.8, 60.9, 65.1, and 55.6 mA h g^−1^ in sequence (Figure 5a). Of more importance, the coulombic efficiency (CE) of batteries operating with LN-SDS-n (n ≥ 1) was greatly increased (Figure 5a, inset), suggesting more reversible electrode reactions [10]. Additionally, the batteries operating with LN-SDS-60 and LN-SDS-90 manifested superior rate capability and electrochemical stability, as they behaved well at 5 C and their *C*_dis_ can resume the initial value after 50 cycles at varied C rates (Figure 5b). Although LN-SDS-n (n = 1, 30, 120) performed better than LN-SDS-0, they cannot endow NTP‖LMO with good rate capability and/or cycling stability.

Considering that LN-SDS-n (n = 60, 90, 120) render NTP‖LMO with high-capacity retention (Figure 5b), their functions on the cycling performance of NTP‖LMO were evaluated (Figure 5c). With LN-SDS-0, the *C*_dis_ of NTP‖LMO retained 17.1 mA h g^−1^ after 600 cycles at 1 °C, presenting a capacity retention of only 26.9%. With LN-SDS-n (n = 60, 90, 120), however, the rate capability and cycling stability of NTP‖LMO were greatly improved. Among the electrolytes, LN-SDS-90 was the top performer. The *C*_dis_ of NTP‖LMO operating with LN-SDS-90 retained 47.1 mA h g^−1^ after 2000 cycles with varying C rates, presenting a capacity retention of 70.4%. When working at 5 °C for 2000 cycles, a *C*_dis_ of 30.8 mA h g^−1^ was retained, presenting a 76.8% capacity retention and a nearly 100% CE (Figure 5d and inset). The high cycling stability is also reflected by the Nyquist plots (Appendix A). The ohmic resistance (*R*_o_) represented by the intercept at the *x*-axis was 0.84 Ω before cycling and increased by only 0.58 Ω after 2000 cycles at 5 °C. In addition, the charge transfer resistance (*R*_ct_) represented by the semicircle in the high- to medium-frequency region increased by only 6.85 Ω after cycling, according to the fitting results with Zsimpwin software. It should be noted that the most rigorous proof for ARMB stability is the performance at low C rates, rather than the high-rate performance [15,17]. Therefore, the electrochemical stability of NTP‖LMO operating with LN-SDS-90 at 0.2 °C was examined. After 200 cycles, the *C*_dis_ retained 41.0 mA h g^−1^, presenting a 61.8% capacity retention and a nearly 100% CE (Figure 5e). The slow but steady capacity fading suggests that the interfacial chemistry needs to be tailored for more effective electrode protection [17]. In brief, LN-SDS-90 outperformed the other LN-SDS-n (n = 0, 1, 30, 60, 120) with regards to making NTP‖LMO behave well at room temperature.

The specific energy (*E*_battery_, W h kg^−1^) and specific power (*P*_battery_, W kg^−1^) of a battery can be obtained according to the GCD curves (Appendix A) [71]. The *E*_battery_ values at varying C rates were calculated to be ca. 47, 45, 41, 35, and 28 W h kg^−1^ using the total mass of anode and cathode materials, with corresponding *P*_battery_ being 18, 45, 90, 173, and 353 W kg^−1^ (Figure 5f). After 2000 cycles at varying C rates (Figure 5c), the *E*_battery_ and *P*_battery_ values remained at 28 W h kg^−1^ and 90 W kg^−1^, which are superior to or comparable with those (34–43 W h kg^−1^, 70–130 W kg^−1^) of ARMBs without and with cycles of less than 300 (Appendix A) [46,48,72,73,74,75].

### 3.4. NTP‖LMO Operating at Zero and Subzero Temperatures

In view of the relatively high ionic conductivity for LN-SDS-n at zero and subzero temperatures (Table 1), LN-SDS-90 was selected for evaluating the low-temperature performance of NTP‖LMO. When operating at 1 °C and zero temperature (Figure 6a), the *C*_dis_ gradually increased and stabilized at ca. 40 mA h g^−1^, due to the activation of the electrodes. After 5000 cycles with varying C rates from 1 to 5, and then to 1 C, the *C*_dis_ retained 35.2 mA h g^−1^, delivering a nearly 100% CE (Figure 6a and inset) and a high-capacity retention (93.1%). Even operating at −15 °C with varying C rates for 1000 cycles, the *C*_dis_ retained 10.7 mA h g^−1^, showing a nearly 100% CE and a 71.3% capacity retention (Figure 6b and inset). However, NTP‖LMO operating with LN-SDS-0 cannot work at −15 °C, counter-evidencing the anti-freezable ability of LN-SDS-90.

## 4. Conclusions

In summary, a new type of electrolyte, i.e., LN-SDS-n, has been designed that is composed of Li^+^/Na^+^-mixed ions and concentrated SDS. The physicochemical properties of the electrolytes have been systematically characterized, which include the electrochemical stability, spectral characteristic, ionic conductivity, rheological properties, and anti-freezable performance. The scientific contributions of this research include: (i) finding the synergistic effect of Li^+^, Na^+^, SO_4_^2−^, and SDS that endows LN-SDS-n (n ≥ 1) with a lowered melting point, reduced freezable water concentration, suppressed water reactivity, and high ionic conductivity in a wide temperature range, and (ii) proposing a micelle-assisted formation mechanism of directional ion transport channels for the SDS-containing electrolytes with high ionic conductivity, based on the theoretical computation results. Using NTP‖LMO as a probe, the electrolytes were screened. LN-SDS-90 was found to be a top performer that can significantly improve the electrochemical properties of NTP‖LMO ARBs. As a low-cost and ecofriendly electrolyte, LN-SDS-90 is a promising candidate for application in ARBs with an output potential smaller than 2 V.

## Figures and Tables

**Figure 1 nanomaterials-12-01920-f001:**
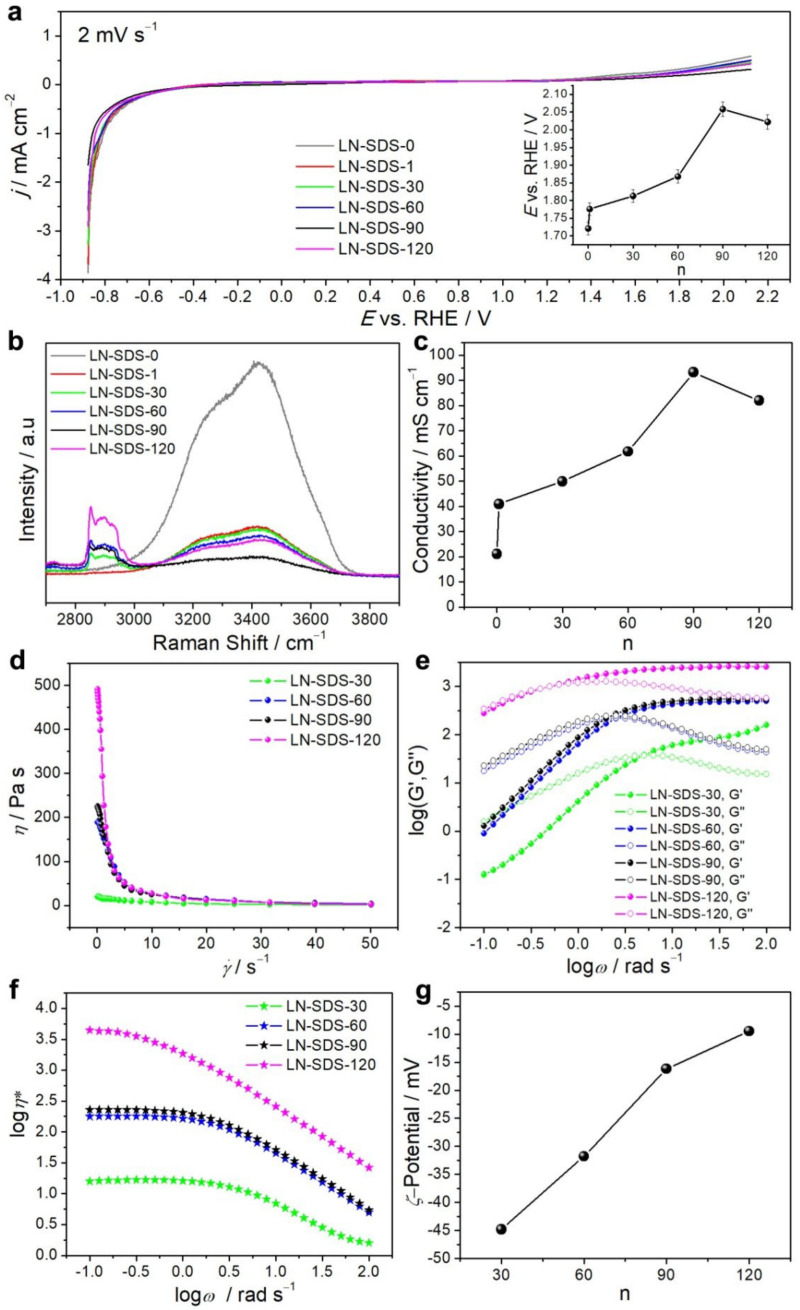
Room-temperature physicochemical properties of LN-SDS-n: (**a**) LSV plots, inset is the ESW plot with relative standard deviation (RSD) of ca. 1%, (**b**) Raman spectra, (**c**) ionic conductivity plot, (**d**) shear viscosity plots vs. shear rate, (**e**) plots of elastic modulus and viscous modulus vs. angular frequency, (**f**) plots of complex dynamic viscosity vs. angular frequency, and (**g**) *ζ*-potential plot.

**Figure 2 nanomaterials-12-01920-f002:**
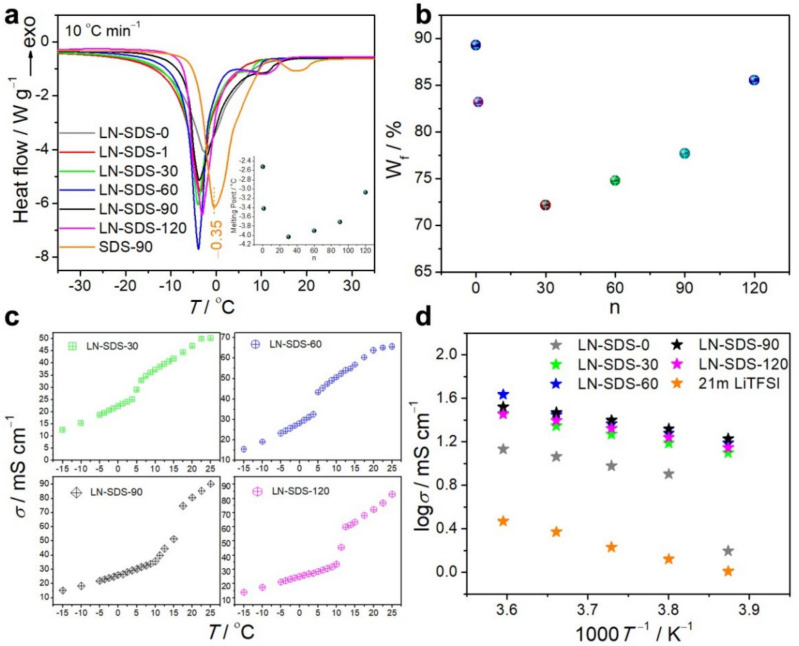
(**a**) DSC curves, inset is the melting point plot (RSD < 0.3%). (**b**) Plot of freezable water concentration vs. n (RSD < 0.3%). (**c**) Plots of ionic conductivity vs. temperature. (**d**) Plots of log *σ* vs. 1000/*T*, and 21 m LiTFSI is provided for comparison.

**Figure 3 nanomaterials-12-01920-f003:**
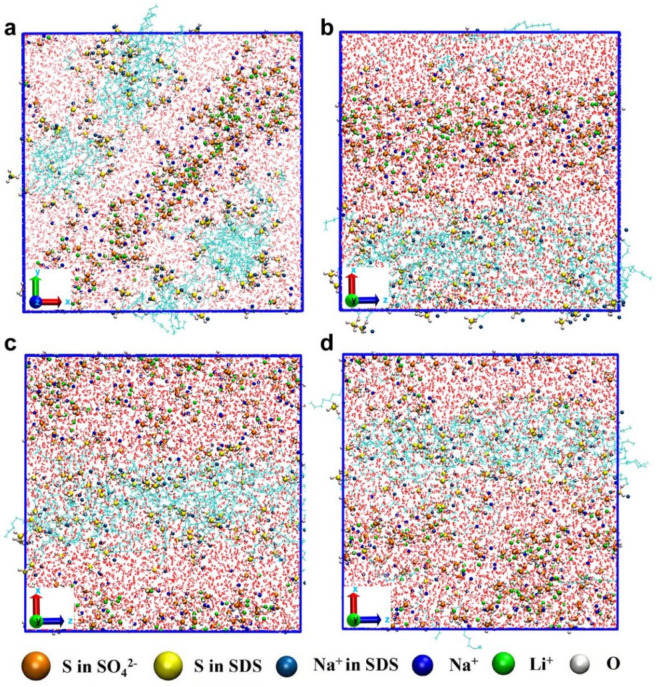
Molecular dynamic simulation results for low-temperature LN-SDS-90 applying a voltage of 2 V nm^−1^ at (**a**) 258.15 K, (**b**) 263.15 K, (**c**) 268.15 K, and (**d**) 273.15 K.

**Figure 4 nanomaterials-12-01920-f004:**
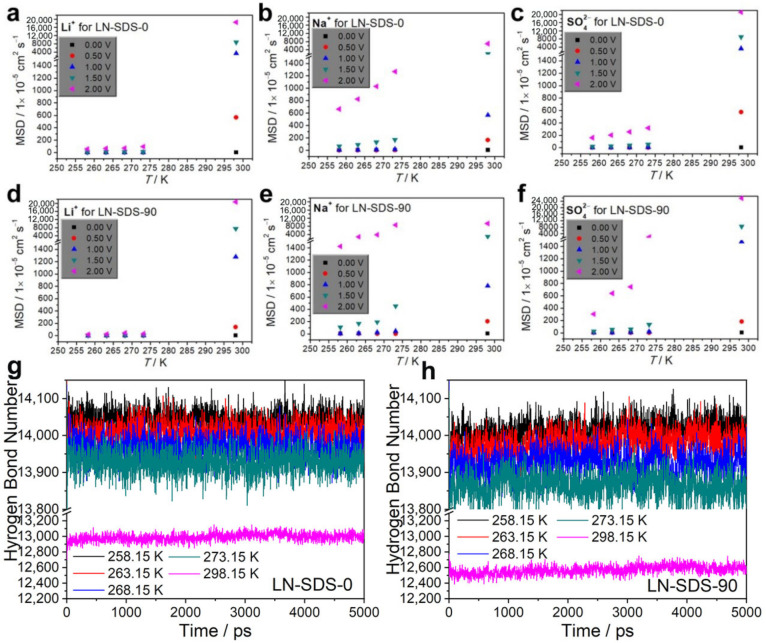
(**a**–**f**) MSD results for LN-SDS-0 and LN-SDS-90 at temperatures of 298.15–258.15 K, without and with applying voltages of 0.5–2 V nm^−1^, and HBNs of water in LN-SDS-0 (**g**) and LN-SDS-90 (**h**) at temperatures of 298.15–258.15 K, applying a voltage of 2 V nm^−1^.

**Figure 5 nanomaterials-12-01920-f005:**
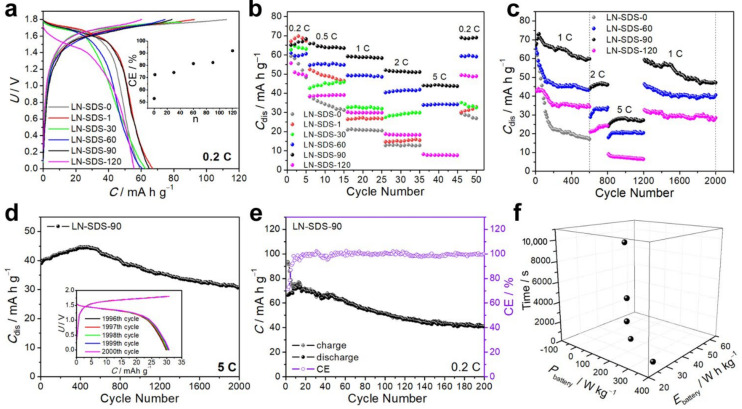
Electrochemical performance of NTP‖LMO operating with LN-SDS-n at room temperature: (**a**) GCD curves, (**b**) rate capability, (**c**) cycling performance at varying C rates, (**d**) cycling performance at 5 C, (**e**) cycling performance at 0.2 C, and (**f**) Ragone plot.

**Figure 6 nanomaterials-12-01920-f006:**
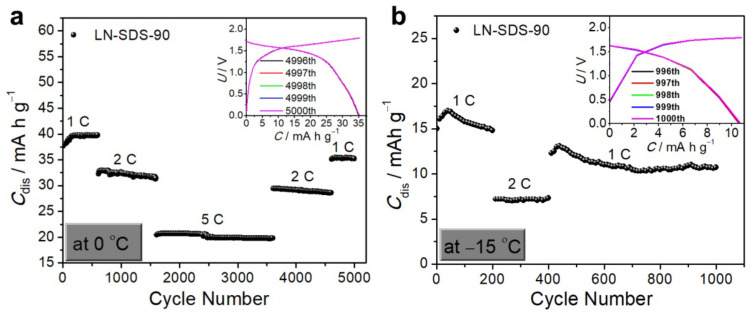
Electrochemical performance of NTP‖LMO operating with LN-SDS-90 at: (**a**) 0 °C and (**b**) −15 °C. Insets in (**a**) and (**b**) are the GCD curves of the final five cycles.

**Table 1 nanomaterials-12-01920-t001:** Comparison of ionic conductivity for LN-SDS-90 and concentrated electrolytes in the literature.

Electrolyte	Conductivity at Different Temperatures (mS cm^−1^)				Ref.
	25 °C/r.t.	10 °C	0 °C	−15 °C	
21 m LiTFSI (WIS)	10, 10.9 ^a^	6.6 ^a^	2.4 ^a^	1.0 ^a^	[17]
Li(TFSI)_0.7_(BETI)_0.3_·2H_2_O (WIBS)	3 at 30 °C	–	–	–	[18]
7 m NaOTf in H_2_O + 8 m NaOTf in PC	25 at 20 °C	–	–	–	[22]
15–35 m NaFSI (WIS)	8–90 at 20 °C	–	–	–	[23]
27 m KOAc	31.4	–	–	–	[25]
32 m KOAc–8 m LiOAc	5.3	–	–	–	[25]
10–30 m KOAc	32–90	–	–	–	[26]
10–40 m HCOOK	46–130	–	–	–	[26]
50 wt% LiPAA gel	6.5 at 20 °C	–	–	–	[27]
25 m LiNO_3_ in water (WIS)	73.8	0.016	–	–	[28]
25 m LiNO_3_ in H_2_O:PD	22.8	14.4	–	–	[28]
low WIS gel	16.2	–	–	–	[28]
2 m LiTFSI–*x*PEG–(1–*x*)H_2_O	0.8–3.4 (*x* = 71–94%)	–	–	–	[8]
polyacrylate derivatives with K^+^ (WIPSE)	45–87				[30]
potassium polyacrylate (WIPSE)	40–120				[31]
organic carbonate-based Li^+^ electrolytes	4.5–10.7	–	–	–	[32]
LN-SDS-90	93.2	35.8	26.1	15.2	this work
LN-SDS-120	82.0	33.7	24.9	14.0	this work

^a^ Denotes the conductivity measured in this work.

**Table 2 nanomaterials-12-01920-t002:** *σ* and *E*_a_ values of the electrolytes at different temperatures.

Electrolytes	*σ* at Different Temperatures (K)/mS cm^−1^							*E*_a_/eV
	298	283	278	273	268	263	258	
LN-SDS -0	21.1	15.6	13.6	11.6	9.5	8	1.6	0.587
LN-SDS-30	50.1	37.2	29	22.3	18.7	15.4	12.6	0.252
LN-SDS-60	65.7	50.6	43.2	28.1	23.3	19.1	15.5	0.242
LN-SDS-90	87.2	60.7	33.2	29.5	25	20.8	16.9	0.211
LN-SDS-120	82.8	45.5	28.5	24.9	21.1	17.4	14	0.221
21 m LITFSI *	10.9	6.6	2.9	2.3	1.7	1.3	1	0.334

* 21 m LITFSI is provided for comparison.

**Table 3 nanomaterials-12-01920-t003:** HBNs of water in electrolytes in the temperature range of 258.15–298.15 K.

Temperature (K)	HBNs of Water in Electrolytes	
	LN-SDS-0	LN-SDS-90
258.15	14,050 ± 50	14,020 ± 50
263.15	14,020 ± 50	13,990 ± 50
268.15	13,960 ± 50	13,920 ± 60
273.15	13,930 ± 50	13,850 ± 50
298.15	12,980 ± 100	12,570 ± 150

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
