# Peer review of "Electrolytes with Micelle-Assisted Formation of Directional Ion Transport Channels for Aqueous Rechargeable Batteries with Impressive Performance"

_nanomaterials, 2022, doi:10.3390/nano12111920_

Round 1

Reviewer 1 Report

This paper deals with the design and characterization of Li2SO4‒Na2SO4‒sodium dodecyl sulfate (LN-SDS)-based aqueous electrolytes, which provide decrease of the water reactivity and increase of the ionic conductivity at the same time. This is well written and organized paper. It is scientifically sound and contains sufficient interest. Few comments are as follows:

1.       (p.2) please define ESW acronym.

2.       (p.3) Wavelength of the X-ray source is missing.

3.       Table 1 (first column, last lines): replace LN-SDS-90 and LN-SDS-120 for LN-SDS-n and delete footers b) and c)

4.       The caption of figure S1 should be more extensive and self-explanatory. Please provide the experimental configuration (electrodes, sample thickness, temperature, etc.)

5.       (p.6, bottom) please describe the Nyquist plots.

6.       Fig. 1a (LSV curves) what are the bar errors for ESW values?

7.       Provide bar errors for data in Fig. 2b

8.       Fig. 2c: the origin of the step in the curve sigma vs temperature is unclear. Please discuss more deeply.

9.       Results in Fig. 4g-h need more discussion.

10.    The scientific contribution of this work should be described clearly in the conclusion part.

Author Response

For reviewer 1#

This paper deals with the design and characterization of Li2SO4‒Na2SO4‒sodium dodecyl sulfate (LN-SDS)-based aqueous electrolytes, which provide decrease of the water reactivity and increase of the ionic conductivity at the same time. This is well written and organized paper. It is scientifically sound and contains sufficient interest. Few comments are as follows:

  Author reply: These comments are highly appreciated.

  1. (p.2) please define ESW acronym.

Author reply: ESW is the acronym of electrochemical stability window, and was expanded in Manuscript R1 (page 4).

  1. (p.3) Wavelength of the X-ray source is missing.

Author reply: Wavelength of the X-ray source is 1.5406 Å, and was supplemented in Manuscript R1 (page 6).

  1. Table 1 (first column, last lines): replace LN-SDS-90 and LN-SDS-120 for LN-SDS-n and delete footers b) and c)

Author reply: Following the reviewer’s direction, LN-SDS-90 and LN-SDS-120 were placed in Table 1, respectively; and footers b and c were deleted in Manuscript R1 (pages 10-11).

  1. The caption of figure S1 should be more extensive and self-explanatory. Please provide the experimental configuration (electrodes, sample thickness, temperature, etc.)

Author reply: This comment is instructive. Following the reviewer’s suggestion, the experimental conditions for EIS measurements were supplemented in the Supplementary data (page S3).

  1. (p.6, bottom) please describe the Nyquist plots.

Author reply: Following the reviewer’s suggestion, description on the Nyquist plots was provided in Manuscript R1 (page 24-25).

  1. Fig. 1a (LSV curves) what are the bar errors for ESW values?

Author reply: The highest relative standard deviation (RSD) for the measured ESW values is ca. 1%. We added the error bars in Fig. 1a (inset) of Manuscript R1 (page 12).

  1. Provide bar errors for data in Fig. 2b.

Author reply: The RSD values for the data in Fig. 2b and the data in Fig. 2a (inset) are smaller than 0.3%. We added the error bars in Fig. 2b and Fig. 2a (inset) in Manuscript R1 (page 15).

  1. Fig. 2c: the origin of the step in the curve sigma vs temperature is unclear. Please discuss more deeply.

Author reply: The origin of the abrupt increase of ionic conductivity (σ) vs temperature is characteristic of surfactant solutions, and is attributed to the Krafft temperature (TK), above which the conductivity suddenly rises due to the increase in surfactant solubility (the blue sentence in page 5 and the revision in page 16).

  1. Results in Fig. 4g-h need more discussion.

Author reply: Following the reviewer’s direction, more discussion was conducted on Fig. 4g and h in Manuscript R1 (pages 21-22).

  1. The scientific contribution of this work should be described clearly in the conclusion part.

Author reply: This comment is very instructive. Following the reviewer’s direction, the conclusion was rewritten to clearly describe the scientific contribution of this research (page 27).

All the revisions are marked in red color in Manuscript R1.

Finally, thanks again for completing reading the above response, and we are looking forward to hearing from you.

Best regards

The authors

Reviewer 2 Report

Thanks for the invitation to review the submitted manuscript which can be published but require major revision which require major revision before publication

1. Authors should include water in polymer salt electrolyte in Table 1 and compare their electrolyte conductivity with them as well such as https://doi.org/10.1016/j.jpowsour.2022.231103; https://doi.org/10.1002/aesr.202100165 etc.

2. The methodology to measure ESW is poor authors should try to use microelectrode to estimate the correct ESW.

3. Authors hsould provide the activation energy using conductivity vs tem plot and explain its physical significance.

4. Why the electrolyte is operable at low temperature is not clear.

Author Response

For reviewer 2#

Thanks for the invitation to review the submitted manuscript which can be published but require major before publication.

Author reply: Thanks.

  1. Authors should include water in polymer salt electrolyte in Table 1 and compare their electrolyte conductivity with them as well such as https://doi.org/10.1016/j.jpowsour.2022.231103; https://doi.org/10.1002/aesr.202100165 etc.

Author reply: The recommended references (refs. 30,31 in the text) were cited and comparatively discussed in Manuscript R1 (pages 4 and 11).

  1. The methodology to measure ESW is poor authors should try to use microelectrode to estimate the correct ESW.

  Author reply: The methodology to measure ESW is according to literature (Nat. Mater. 19 (2020) 1006-1011). Measurements by microelectrode may be a promising method in evaluating the electrochemical properties of electrode materials for EES, and we will try it in future research.

  1. Authors should provide the activation energy using conductivity vs tem plot and explain its physical significance.

Author reply: This comment is instructive. Following the reviewer’s direction, we supplemented results and discussion on the activation energy of electrolytes through the Arrhenius equation in Manuscript R1 (pages 17-18).

  1. Why the electrolyte is operable at low temperature is not clear.

  Author reply: Based on the analysis of activation energy by Arrhenius equation, the reason for LN-SDS-n (n = 30–120) operating at low temperatures can be deeply understood (pages 17-18). In brief, (1) LN-SDS-n (n = 30–120) cannot be completely solidified even at –15 °C, because the electrolyte conductivities obey the Arrhenius law; the Arrhenius relationship means the ion motion is liquid-like [67] (page 17); (ii) the higher ionic conductivity of LN-SDS-n (n = 30–120) compared with LN-SDS-0 and 21 m LITFSI is attributed to the lowered Ea, which results from the liquid-like ion motion due to the anti-freezable ability (page 18).

All the revisions are marked in red color in Manuscript R1.

Finally, thanks again for completing reading the above response, and we are looking forward to hearing from you.

Best regards

The authors

Round 2

Reviewer 2 Report

Authors have revised the manuscript well therefore, it can published in its current form.